# A New Approach for Including Social Conventions into Social Robots Navigation by Using Polygonal Triangulation and Group Asymmetric Gaussian Functions

**DOI:** 10.3390/s22124602

**Published:** 2022-06-18

**Authors:** Raphaell Maciel de Sousa, Dennis Barrios-Aranibar, Jose Diaz-Amado, Raquel E. Patiño-Escarcina, Roque Mendes Prado Trindade

**Affiliations:** 1Instituto Federal da Paraíba (IFPB), Campus Cajazeiras, Cajazeiras 58900-000, PB, Brazil; 2Electrical and Electronic Engineering Department, Universidad Católica San Pablo, Arequipa 04001, Peru; dbarrios@ucsp.edu.pe (D.B.-A.); jose_diaz@ifba.edu.br (J.D.-A.); rpatino@ucsp.edu.pe (R.E.P.-E.); 3Instituto Federal da Bahia (IFBA), Campus Vitória da Conquista, Vitória da Conquista 45078-300, BA, Brazil; 4Department of Technologics and Exacts Sciences, State University of Southwest Bahia (UESB), Vitória da Conquista 45083-900, BA, Brazil; roquetrindade@uesb.edu.br

**Keywords:** social robots navigation, social conventions, socialization features, delaunay triangulation, asymmetric Gaussian function

## Abstract

Many authors have been working on approaches that can be applied to social robots to allow a more realistic/comfortable relationship between humans and robots in the same space. This paper proposes a new navigation strategy for social environments by recognizing and considering the social conventions of people and groups. To achieve that, we proposed the application of Delaunay triangulation for connecting people as vertices of a triangle network. Then, we defined a complete asymmetric Gaussian function (for individuals and groups) to decide zones where the robot must avoid passing. Furthermore, a feature generalization scheme called socialization feature was proposed to incorporate perception information that can be used to change the variance of the Gaussian function. Simulation results have been presented to demonstrate that the proposed approach can modify the path according to the perception of the robot compared to a standard A* algorithm.

## 1. Introduction

Robots are special machines with specific objectives and use: for instance, from cleaning and vacuuming households with different layouts to building complex pieces in an industrial environment. In this sense, many researchers have tried to categorize the robots. Gonzalez-Aguirre et al. [1] considered conventional robots and advanced robots working on different tasks and services. These services could include package delivery [2] and catering applications [3], among others [4,5].

Wirtz et al. [6] explain a categorization of service robots in relation to their task type, where a “social robot” could be considered as a type of service robot with emotional–social tasks. However, Duffy et al. [7] are the first that mentioned a “social robot” as a complex entity in an environment that is designed to behave appropriately on its own goals and those of its group of individuals.

According to recent statistics, social robots are more present and commonly seen in our daily lives [8]. When robots share environments with human beings, they often play a role of a friend collaborating with them in specific daily tasks. Examples of such collaboration can be a mobile manipulator, which is designed to serve as an assistant to nurses or in rehabilitation tasks in hospital environments [9], or a mobile robot designed to work as a student receptionist at a university [10], among other robots designed to help in daily human activities.

The interaction among robots and human beings is most of the time inevitable, requiring a further understanding of the social interaction behavior and sentiment that people need to transmit to the robot [11]. In this sense, it is important to acknowledge the environment, known as social space, where the robots interact with humans and other robots. Ewick [12] defines public space as a place inside or outside where people have the opportunity to socialize. Social robots are developed to act in those spaces.

In this direction, it is required to study different scenarios and develop solutions that include human–robot interaction, where psychology and sociology also converge [13]. Thus, many authors have been working to create frameworks that can be applied to social robots for allowing a more realistic/comfortable relationship between humans and robots in the same space. However, little has been known about the different relationship methods (physical, emotional, social, and safe) between human and robots [14].

The idea of creating a “softer” situation of co-existence is not an easy task in a real-life situation because it could bring about infinite and unpredictable situations. Indeed, to socially interact with humans, a robotic system should be able to understand a user’s behavior and intentions and estimate the emotional state with a high level of acceptability and usability [15]. Nevertheless, more attention should be drawn to assessing the feasibility of a social robot in real-life scenarios among different cultures and people [16].

The contributions of this work can be summarized as follows: (1) a new form of recognizing groups by using Delaunay triangulation which includes social conventions, (2) asymmetric Gaussian functions applied to model proxemics in persons and groups, (3) proxemics zones of groups treated as a unique one, not as a composition of individuals zones, (4) the definition of social features to model social conventions and human characteristics, and (5) a process to permit the change of Gaussian functions, of groups and individuals, based on the composition of values of social features.

The paper is structured as follows: Section 2 presents a preliminary introduction of related works regarding path planning strategies with a social approach. Section 3 describes materials and methods for the proposed algorithm. Section 4 presents some simulations results to validate the proposed approach. Section 5 presents a discussion about path planning, taking into account social constraints. Finally, Section 6 presents conclusions.

## 2. Related Work

Sehested et al. [17] in 2010 talked about the problem of fusing socially acceptable behaviors into robot path planning. Sehested explained a specific behavior of a robot where he needs to avoid the paths through work stations to avoid distractions to colleagues. Garrell et al. [18] presented a wide study about the cooperation and collaboration of teams of robots that work cooperatively to develop a specific task. Gomez et al. [19] developed a mathematical formulation for the socially acceptable path planning problem, which defined a Euclidean space with obstacles that could be static or dynamic. These dynamic obstacles, as people, have properties: velocity or directions. Other works developed different social features that could be considered for path planning such as [20,21,22,23].

In a social environment, if a robot needs to interact with people, it needs to respect a specific distance so that everyone could feel secure. In this sense, many approaches that proposed navigation in social environments used proxemics definition and included social features that a robot needs to consider for path planning. In other words, as Glass et al. said, navigation could be influenced by the application. So, Glass et al. [24] proposed navigation strategies for spaces as shopping centers; also, Kanda et al. [25] explained the challenges of public spaces navigation of autonomous robots, and Sehested et al. [17] developed experiments in a work office.

When there is an interaction between robots and humans in a real-world situation, it is expected that the robot can understand and store information about the environment, which is provided by the context in which it was inserted. Some works have considered different features for this problem since human states or even cultural factors can be used as data for social navigation [26]. In this scenario, some approaches aim to extract features from the environment and use them for perception [27]. Zaraki et al. [27] presented an external multi-platform system to extract resources that can be applied to a social robot perception system. In this case, they proposed a perception system based on modularity, inter-connectivity, extendability, and communication. The proposed system had four layers: data acquisition, high-level features extraction, meta-scene creation, and communication. Practically, all models for social interaction between humans and robots, presented there, used various features extracted from the environment, and the collected data were used to provide knowledge about the surroundings. In Vega et al. [28], the authors proposed a trajectory planning system that considered the time of day and the possibility of not using some spaces in specific periods of time. The authors proposed a method that restricts or penalizes the route planned by the robot due to time-dependent variables.

Modeling a dynamic social space is a complex task and involves several variables. Pandey et al. [29] proposed an algorithm that treats humans not as obstacles to be avoided, but rather, the authors defined a social space around a person as a parametric region. This parametric region introduced boundaries around people based on proxemics concepts. The proposed technique was not generalized to groups of people. Later, the idea of parametric regions was studied in depth by Truong et al. [30]. Xuang et al. introduced the concept of Dynamic Social Zones (DSZ), which is a merge between the extender personal space and the social interaction space. The authors proposed to incorporate their approach into the cost function of the D* [31]. Comfort indices that evaluate the proposed method were presented and compared to the D* path planner [32] without social restrictions. Moreover, that paper introduces the application of Gaussian functions to cover the DSZ space and delimit the boundaries of that area.

In [33], the authors expanded their previous work [30] and continued studying DSZ to introduce a module able to predict poses to incorporate group interaction and people in different positions or features, such as a sitting person, a moving person, interaction with some object, and gaze direction. In Chen et al. [21], the authors used the A* algorithm [34] to avoid obstacles and introduced the use of an asymmetric Gaussian function to define a social zone [35]. The idea incorporates asymmetry while delimiting the boundaries of the Gaussian function. Asymmetry was introduced by using the proxemics relationship to modify the variances of the Gaussian function. Simulations and experimental results were presented. On the other hand, the authors did not use other features in addition to proxemics to modify variances, and the proposed technique was not used to generalize different groups of people. Another observation is that the results of the evaluation of the proposed method were not presented, so even though the use of an asymmetric Gaussian function seems to be a reasonable approach, its contribution was not clear compared to Truong [30].

In [36], the authors proposed an autonomous navigation system in a crowded environment, which was named the proactive social motion model (PSMM). The proposed method uses some human states such as position, orientation, motion, field of view, hand pose, and the human–object and human group interactions. Although the proposed method is applied to crowded environments, the authors presented experimental results for a small group of people. They stated that in a chaotic environment, the computational cost for extracting features in real time would affect the data processing of the proposed method. Another approach to model the social space was proposed by [37]. The authors have proposed a new definition of social space, named Dynamic Social Force (DSF). This new social space definition is based on a fuzzy inference system, and the parameters of these functions are adjusted by using reinforcement learning. The authors used reinforcement learning to define the parameters for the Gaussian function. This method was not investigated further as a method to generalize groups of people.

The approach presented in this paper is unique; however, it carries out some familiarities to the work of Gines et al. [38], who proposed the use of dynamic proxemics zones modeled as a circle by modifying the amplitude of Gaussians when recognizing the person’s state in the robot’s environment.

## 3. Materials and Methods

Human beings are social individuals; they live in groups, and throughout history, different societies have developed their own co-existence rules based on political, economic, and cultural conventions [16]. In a social context, the main idea is introducing to autonomous robot navigation the perception of interaction between people and how these people interact with a social environment. Furthermore, it is important to investigate the effects of the robot perception under this social context and how this affects path planning approaches. To exemplify, consider a simple case (see Figure 1). In this Figure, the path planning algorithm (e.g., A* Algorithm) calculated the shortest path between the origin and the goal (path 1), but to reach this goal, the robot must pass between these people. When considering social conventions, this trajectory is not acceptable for humans; so, why should someone expect such behavior from robots? In this case, it is reasonable to think that the robot, like people, should be able to understand social conventions and proxemics relationships and consider that in the path planning algorithm, the robot should not only take the shortest path but also choose an optimized path considering the role of social context (path 2).

In a social situation, an autonomous mobile robot must interact among people or groups of people. As previously presented, the robot must follow social conventions and plan its navigation according to the environment in which it is inserted. Nevertheless, understanding the surrounding environment and classifying a group of people is part of a more realistic experience in terms of social perception for an autonomous robot. As presented in [39], F-formations were used to classify groups of people. The problem with this approach and other ones, as presented in [30,36], is that in a real chaotic environment, where people are grouped in random positions, it is hard to identify any patterns, as shown in Figure 2, with the number of people n=50. The situation presented in Figure 2 makes the inclusion of social conventions in robot behavior difficult, because in real environments, robots will encounter and must respect social conventions with individuals and groups of people.

To enhance the limitations presented in chaotic environments, this paper proposes a new navigation strategy for social environments by recognizing and considering the social conventions of persons and groups. For this, first, it was proposed to recognize groups by using Delaunay triangulation; next, the so-called “social features” for modeling social conventions were defined, which are used in persons and groups to influence proxemic spaces and also robot navigation. The proposed algorithm connects everyone into the space of the robot as vertices of triangles for creating a mesh of triangles (polygonal triangulation). Then, aiming to identify groups of people, this approach acts into social space and uses features such as proxemics and person orientation. The result is the composition of groups of people in a social environment. This approach can generalize the space zone for a group of people and may help the robot learn how to recognize these different patterns of social groups. After all, the medium center of mass of each individual triangle is used to calculate the group center of mass; then, we used an asymmetric Gaussian function from that point to cover the resulting group.

Next, each block of the proposed system will be detailed.

### 3.1. Social Awareness Path Planning Approach

Figure 3 presents the proposed method for path planning with social awareness applied to mobile robots. The proposed approach aims to solve path planning issues with social awareness in an automated and dynamic way. It is based on the use of asymmetric Gaussian functions for modeling proxemic zones to “modify” the robot’s perception of the environment; then, the robot can avoid people based on social conventions and perceptions. The novelty of this approach will be better described subsequently.

The novelty of the proposed approach is next:Asymmetric Gaussian functions can be modified dynamically by using socialization features to change the variance value on each side of the sagittal and frontal axis, as explained in Section 3.3.3.Groups of people are recognized by using polygonal triangulation, which includes proxemic rules and people-orientation to determine what a group is constituted of, as explained in Section 3.2.Each group (recognized using polygonal triangulation) or person alone has its own asymmetric Gaussian functions; individual Gaussian functions are not merged, so a group is considered as an individual with its own proxemics rules. In addition, social features are inherited from humans who are part of each group) to groups.Any amount of socialization features with different types of influence on each side of the Gaussian function could be considered, so any form of modeling the personal space, and also any type of approach for modifying a Gaussian function based on cultural, social, personal, etc., characteristics could be applied.

Therefore, summing up as shown in the proposed block diagram (see Figure 3), all perception acquired from sensors is sent to the social path planning approach. Then, a features extraction block must be implemented to extract relevant features. Then, the data highlighted in this paper can be used to generate a map deriving from the environment. People’s position and orientation on the map are going to be used for triangulation (*Groups Recognition block*). As presented earlier, this block will classify groups of people. Then, the result of triangulation will be added to a cost map with a Gaussian block (*Asymmetric Gaussian with Features* block), which in turn will calculate the parameters of the asymmetric Gaussian function that will be finally applied to the group classification. So, the provided path planning method must include the values of Gaussian functions as a cost function to generate a path to the destination, including people’s awareness based on social conventions. The robot’s resultant path should be used by the robot to navigate, avoid obstacles, and socially drive the robot to the target.

### 3.2. Groups Recognition

The first step in the proposed approach is groups recognition consisting of classifying a group of people in a chaotic situation. First, Delaunay triangulation is applied to connect people as vertices of a single figure, as presented in Figure 4. It is possible to notice that each person represents a connection at the edge of triangles. Consequently, increasing the number of people will lead to a greater number of triangles. In addition, it is possible to see in Figure 4 that each person has a vector that represents the associated direction.

After applying triangulation, the idea is to set two rules aiming to remove triangles so that some connections can be slowly eliminated in a way that the resulting figure will be formed by small parts of the previous singles mesh grid (Figure 4); these resulting parts refer to the groups of people.

The first rule will be characterized here by the distance between each connecting point or the proxemics distance. To calculate the distance, we considered the proxemics relationship between the vertices of a triangle, as presented in Figure 5. Equation (Equation 1) is used to calculate the distance from one point to another.
(1)$$d=\sqrt{{\left(x_{2}-x_{1}\right)}^{2}+{\left(y_{2}-y_{1}\right)}^{2}}$$
where $x_{1},\;x_{2},\;y_{1}$ and $y_{2}$ are coordinates of people positions, and *d* is the Euclidean distance. Figure 5 presents the distance between vertices calculated by Equation (Equation 1).

At this point, the second rule, regarding orientation vectors, will be introduced. Consider that each person can be identified in a 2D space by their position (x,y coordinates) and orientation (θ) as it follows:(2)$$P_{i}(x_{i},y_{i},|v_{i}\left|,\theta_{i}\right)$$
where $|v_{i}|$ is the module of the orientation vector $v_{i}$, which will be defined later, and the label *i* refers to the number of each person.

As presented in [39], orientation is important to model the relationship between people; there is an o-space region, a convex empty space that is surrounded by people who are involved in a social interaction, where the vector *v* and θ should point to this region. In the triangulation context, the proposed o-region is the area inside of the triangle (see the yellow region, in Figure 5), which is named here as *T-space*.

As a prerequisite to form a group, all vectors must be inside of this triangle. Let us consider, then, the region limited by the triangle as a closed set. It will be defined here that all vectors must be contained inside of this set. Now, consider that the extremity of the direction vector of a person has as coordinates the point $P_{v}\left(x_{v},y_{v}\right)$, $P_{i}\left(x_{i},y_{i}\right)$; this point must be contained inside of the *T-space* as follows:

The superscript $v_{i}$ refers to the vector, and the subscript *i* refers to one of the people.
(3)$$P_{v_{i}}\left(x_{i},y_{i}\right)\subset T$$
where the superscript $v_{i}$ refers to the vector, and the subscript *i* refers to one of the people that could make up a triangle.

The triangle defined by *T-space* (see yellow triangle in the Figure 5) will be the reference one ($T^{*}$) for a single group. It is necessary to define a rule to include peripheral triangles since they have a single vector inside their *T* region. Still, two points are included in the $T^{*}$. This situation is presented in Figure 6 and Figure 7.

Figure 6 presents a reference triangle ($T^{*}$) and another peripheral $T^{´}$. However, it is possible to see that the orientation vector of the person relative to the peripheral triangle is not pointing to $T^{*}$. In this case, this peripheral triangle will not be included in this group.

In this scenario shown in Figure 7, the person’s orientation from the peripheral triangle has been changed. It is now pointing to $T^{*}$. The second rule is going to apply to every peripheral triangle.

It is possible to notice that for crowded environments, represented as everyday situations, a ride on a mall, for example, it is relatively easy to apply Delaunay triangulation for connecting people as vertices of triangles. However, triangulation creates a single mash figure, and to classify groups, and it is necessary to use more information. Then, we proposed here a new space definition that uses a *T-space* to define a zone where people who have social interaction must share the space. It represents a simple situation for a single person or two people where the regular proxemics rules can be applied [30].

### 3.3. Features for Social Awareness

When a robot is introduced into a social environment among people, mirroring behavior is expected from the robot in every detail, including demonstration of social awareness. Since social awareness is a human ability, it is natural to show empathy with others, but robots do not pursue this ability. Therefore, it is important to develop algorithms that allow robots to demonstrate it. This way, when humans interact with robots, humans can feel comfortable with the robots’ presence.

When thinking about social interaction spaces, the implementation of social awareness behavior in robots must respect the perceived range of each proxemic distance (intimate, personal, social, and public) to every person that the robot is interacting with. That modification must be based on the perception and features extracted from the situation to understand the social conventions in the surrounding environment and the necessities of human beings [26,27].

Social conventions and necessities of humans in a social environment can be modeled as *socialization features*, which are quantitative measures of information related to the expressions of social acceptance of humans. Those expressions can be based on cultural, emotional, and contextual characteristics of people and the environment at a specific time.

In this sense, the second step of the proposed approach is the use of socialization features for changing the values of the Gaussian function into the sagittal and frontal axis of a person in an asymmetric way and also extending this behavior to groups of people interacting between them; for that, next, we will formalize the definition of socialization features, the asymmetric Gaussian function, and its adaptation by using the former ones.

#### 3.3.1. Socialization Features

When designing robots with social capabilities, it is important to apply user modeling, a quantitative or qualitative evaluation of parameters, attributes, or metrics of many types such as cognitive, attentional, etc., for describing a user or group of users. Those models can be employed for many purposes such as understanding humans, controlling feedback, or adapting a robot’s behavior [40].

It is proposed to use numerical vectors called *socialization features* to act as users’ social conventions modelers, considering that all humans are potential users of the robotic system in a social environment.

Applying modeling in social environments could also be understood as robot acceptance models. Those models are widely studied in the assistive social robots scope and could include features such as anxiety (how anxious is a person when interacting with the robot), attitude (a person has positive or negative feelings about a robot), intention to use (the intention of using the robot is positive or negative), perceived enjoyment (the user experiences joy or pleasure by interacting with the system), social influence (users near the user have a positive or negative attitude to the robot), among others [41,42,43]. There are a lot of parameters and perceptions that can be used as a source for social models. For example, when talking about emotions and feelings, it is possible to identify the facial expressions to determine if a person is angry, disgusted, fearful, happy, neutral, sad, or surprised [44]. In addition, it is possible to make speech emotion recognition for the same purpose: recognizing emotions such as anger, fear, happiness, neutral, sad, and surprise in humans by their way of speaking [45].

In addition, the aforementioned parameters called “social cues” [46] are verbal and non-verbal messages that influence social signals that affect socialization actions such as approaching or not to others. Those social cues could include: Physical appearance (Height, Attractiveness, and Body shape), Gesture and posture (Hand gestures, Posture, and Walking), Face and eyes behavior (Facial expressions, Gaze behavior, and Focus of attention), Vocal behavior (Prosody, Turn-taking, Vocalizations, Silence), and Space and Environment (Distance and Seating arrangement).

It is proposed to express *socialization features* as a vector $v_{i}=\left[v_{1},v_{2},v_{3},\ldots ,v_{k}\right]$ where $v\in R^{k},\;v_{j}\in R,$
$0<v_{j}<=1$ for all j∈N and 0<j≤k, *k* is the number of levels of the feature and *i* is the order number of the feature in the set of features used for a social robot application. Each feature must represent a range of possible values of a specific social cue; also, each numeric value must represent the level of positiveness or negativeness of the observed feature in a person. For example, let us consider the feature “*emotional state*” shown in Table 1 with seven possible values.

In Table 1, higher values of a $v_{j}$ refer to a more socially acceptable social expression; in other words, it indicates that a person is more likely to be closer to the robot; therefore, it is possible to plan a path considering a small boundary around the identified person.

#### 3.3.2. Asymmetric Gaussian Function

Social interaction spaces are widely modeled by using Gaussian functions [21,30,47,48]. So, since spaces occupied by humans and robots can be modeled as two-dimensional ones, it is common to model those spaces with two-dimensional Gaussian functions, which are expressed as follows:(4)$$f\left(x,y\right)=e^{-\left(\frac{{\left(x-x_{c}\right)}^{2}}{2\sigma_{x}^{2}}+\frac{{\left(y-y_{c}\right)}^{2}}{2\sigma_{y}^{2}}\right)}$$
where $\left(x_{c},y_{c}\right)$ denotes the center of the Gaussian (the position of the person), (x,y) denotes a point where the value of Gaussian function will be calculated, $\sigma_{x}^{2}$ is the variance in the *x* axis, and $\sigma_{y}^{2}$ is the variance in the *y* axis. In addition, it is important to note that this two-dimensional Gaussian is symmetric on each axis (*x* and *y*).

In real social environments, interaction spaces are not symmetric. Some reasons, such as social interaction among groups of people, can cause a lengthening of the intimate space forward, cultural aspects can cause the intimate space on one side to be larger than the other, and so on. In this sense, it is important to model the social interaction spaces by using asymmetric Gaussian functions. Several approaches were proposed for constructing asymmetric functions, such as that proposed by Chen et al. [21]. The proposal considered that the *x* and *y*-axis are aligned with the frontal and sagittal axis of the human body, respectively. They defined the left side and right side variances for modifying the variance along the axis (frontal axis) and the front and rear variances for modifying the variance along the *y* axis (sagittal axis).

Since a human body cannot always be aligned to the *x* and *y* axis, it is important to consider the person’s orientation in the equation for calculating the asymmetric Gaussian function. Rachel Kirby [47] made a proposal based on this fact; however, she only proposed an asymmetry in the sagittal axis (front and rear variances) and considered the frontal axis to be symmetric (left side and right side variances were considered the same).

This work proposes using a completely asymmetric Gaussian function to model interaction spaces (such as the work of Chen et al. [21]) considering that the frontal and sagittal axis is rotated θ degrees in relation to *x* and *y* axis (resembling the work of Rachel Kirby [47]). θ is the angle that defines the person’s orientation related to a reference frame of the world that humans and robots share. Figure 8 shows the basis of the proposed approach.

Given a point (x,y) in the world’s reference frame, it is important to calculate the variance along the sagittal and frontal axis. Thus, as shown in Figure 9 and Figure 10, the distance in the sagittal axis can be calculated by adding the values of segment *a* and *b*, and the distance in the frontal axis can be calculated by adding the values of segments *c* and *d*, where $a=cos\left(\theta \right)\left(x-x_{c}\right)$, $b=sin\left(\theta \right)\left(y-y_{c}\right)$, $c=sin\left(\theta \right)\left(x-x_{c}\right)$, and $d=-cos\left(\theta \right)\left(y-y_{c}\right)$. So, the two-dimensional Gaussian function that considers orientation (θ) of the human in the reference frame of the world can be expressed as follows:(5)$$f\left(x,y\right)=e^{-\left(\frac{{\left(d_{sag}\right)}^{2}}{2\sigma^{2}}+\frac{{\left(d_{fro}\right)}^{2}}{2\sigma_{s}^{2}}\right)}$$
where $d_{sag}=cos\left(\theta \right)\left(x-x_{c}\right)+sin\left(\theta \right)\left(y-y_{c}\right)$, $d_{fro}=sin\left(\theta \right)\left(x-x_{c}\right)-cos\left(\theta \right)\left(y-y_{c}\right)$, $\sigma^{2}$ is the variance along the sagittal axis, and $\sigma_{s}^{2}$ is the variance along the frontal axis.

After applying a set of basic mathematical properties, the two-dimensional Gaussian function that considers orientation (θ) of the human in the reference frame of the world can be expressed as follows (such as the work of RachelKirby [47]):(6)$$f\left(x,y\right)=e^{-\left(g{\left(x-x_{c}\right)}^{2}+2h\left(x-x_{c}\right)\left(y-y_{c}\right)+k{\left(y-y_{c}\right)}^{2}\right)}$$
where
$$g=\frac{{\left(cos\left(\theta \right)\right)}^{2}}{2\sigma^{2}}+\frac{{\left(sin\left(\theta \right)\right)}^{2}}{2\sigma_{s}^{2}},$$
$$h=\frac{sin(2\theta )}{4\sigma^{2}}-\frac{sin(2\theta )}{4\sigma_{s}^{2}},\;and$$
$$k=\frac{{\left(sin\left(\theta \right)\right)}^{2}}{2\sigma^{2}}+\frac{{\left(cos\left(\theta \right)\right)}^{2}}{2\sigma_{s}^{2}}$$

Finally, to have a completely asymmetric Gaussian function, it is important to define four variances: $\sigma_{f}$, $\sigma_{r}$, $\sigma_{ri}$, and $\sigma_{l}$ for the front, rear, right and left sides of humans, respectively. So, σ and $\sigma_{s}$ vary according to the relation between the angle formed by the point (x,y) (where the value of the Gaussian function will be calculated) and the orientation of the human in the reference frame of the world. That relation is modeled in a so-called angle α, and it can be expressed as follows:(7)$$\alpha =atan2(y-y_{c},x-x_{c})-\theta +\pi /2$$

So, σ takes values of $\sigma_{f}$ and $\sigma_{r}$, and $\sigma_{s}$ takes values of $\sigma_{ri}$ and $\sigma_{l}$, according to the next equation:(8)$$\left(\sigma ,\sigma_{s}\right)=\left\{\begin{array}{c} (\sigma_{f},\sigma_{ri})\;\mathrm{if}\;0\leq \alpha <\pi /2\\ (\sigma_{f},\sigma_{l})\;\mathrm{if}\;\pi /2\leq \alpha <\pi \\ (\sigma_{r},\sigma_{ri})\;\mathrm{if}\;-\pi /2\leq \alpha <0\\ (\sigma_{r},\sigma_{l})\;\mathrm{if}\;-\pi \leq \alpha <-\pi /2 \end{array}\right.$$

#### 3.3.3. Adaptive Gaussian Using Features

When talking about personal spaces (where the intrusion of others can cause discomfort), it is important to note that various shapes were proposed [46] (circular, egg shape higher in the front, ellipse, and smaller in the dominant side). So, depending on the situation, the context, the cultural and the emotional characteristics of social interaction, the shape of proxemic zones could change. It is proposed to dynamically modify it depending on such factors modeled as socialization features.

Since proxemic zones are usually modeled using Gaussian functions, in this paper, it is proposed to use the totally asymmetric Gaussian function presented before Equation (Equation 6) with its four variances (Equation (Equation 8)) modified dynamically considering the value of socialization features. For this purpose, it is important to associate all features’ numerical values to each of the four variances by using *influence vectors*.

An *influence vector* is a column vector of real values $\mathrm{IV}={\left[IV_{1},IV_{2},IV_{3},\ldots ,IV_{n}\right]}^{\prime }$ where $\mathrm{IV}\in R^{n},\;IV_{i}\in R,$$-1<=IV_{i}<=1$ for all i∈N and 0<i≤n, and *n* is the number of features considered to influence the proxemic zones. It is important to clarify that there are four independent *influence vectors*: ${\mathrm{IV}}^{f}$, ${\mathrm{IV}}^{r}$, ${\mathrm{IV}}^{ri}$, and ${\mathrm{IV}}^{l}$ for the front, rear, right and left sides of humans, respectively.

Given that influence values for each feature into the *influence vector* are between −1 and 1, it is important to note that positive influence values mean that as the value of the feature increases, it is possible to get closer and negative values mean that as the value approximates −1, it is better to move away from the person or the group of people.

In execution time, socialization features will influence the variance of Gaussian functions with the help of a row vector FV of instant values of features, $FV=[FV_{1},FV_{2},FV_{3},\ldots ,FV_{n}]$ where $\mathrm{FV}\in R^{n},\;FV_{i}\in R,$ $0<FV_{i}<=1$ for all i∈N and 0<i≤n, *n* and 0<i≤n, *n* is the number of features considered to influence the proxemic zones, and $FV_{i}$ is the value that the *i*-th socialization feature takes in one moment during the robot interaction with humans. That former value must be obtained in the perception module of the robotic system.

So, the next step is to calculate the influence value (iv) for each side of the asymmetric Gaussian function. The former value must be used to modify, in a proportional way, the variance of each side of the Gaussian function by multiplying it ($\sigma^{2}=iv*\sigma^{2}$). It is important to note that iv is calculated by applying the dot product between vectors FV and IV (iv=FV·IV).

#### 3.3.4. Adaptive Gaussians in Groups of People

Adaptive Gaussians were used for individuals, and in the case of groups of people, the common approach is to join individual Gaussians for having union between proxemic zones. However, this traditional approach is not the best selection, since when people form a group, the last becomes an entity with its own proxemic rules and zones. So, it is proposed that groups must be considered as individuals, while the personal zone acquires a new semantic in a group, which is the space where the group is only able to interact among good friends or family.

For having a group considered as an individual, it is important to first define its position and orientation; after that, it is important to define the shape and dimensions of the group on the left, right, rear, and front sides; finally, it is important to modify the Gaussian based on features of the group. All those aspects are explained next:Position of the group is the midpoint of all the positions of persons included in the group.Orientation of the group is the mean of orientations of all the individuals who compose the group.The group shape is defined as an irregular quadrilateral, where the four vertices are calculated considering the position and orientation of the group. From the orientation, four quadrants are defined, and on each quadrant, the position of the farthest person from the center of the group is calculated. Each of the four founded positions is labeled as left, right, rear, and front points, depending on the quadrant.The group has one unique feature value, which can be calculated in the perception module of the robot by combining the feature values of individuals by averaging, finding the statistical mode, and so on.

## 4. Results

For proof of concept, the proposed algorithms were implemented in simulated environments using Matlab. Results were performed on a core i7 computer with a Titan Xp graphic card and 16 Gb of memory. This paper is part of a project named RUTAS (Robots for Urban Tourism Centers, Autonomous and Semantic-based), where they used the Pepper robot to navigate in a social context. Pepper has three head tactile sensors, one 3D sensor, two RGB cameras, two infrared sensors, one IMU, and two sonars. So, it was planned to implement the proposed approach in Pepper as a continuance of this project by using its sensors as input for the proposed algorithm. Data from cameras and 3D sensors can be used to detect humans and implement sentiment analysis as input to the emotional state detection (see Figure 3). The results were divided as follows:The proposed groups classification method based on Delaunay triangulation;Inclusion of people orientation in the proposed group classification method;Asymmetric Gaussian application with socialization features to define personal and group zones;The proposed navigation strategy.

### 4.1. Groups Classification

Simulations results will be presented here to demonstrate the proposed group classification strategy. First, Delaunay triangulation was used to connect people as vertices of a triangle mesh, as presented in Figure 11a. In this figure, there is a random people distribution (n=10 points representing people), with distance (*d*) in meters and results in unit percentage (p.u.).

Figure 11b presents the same random distribution presented in Figure 11a with the addition of a maximum distance ($d_{max}=0.40$p.u.) proxemics relationship (Equation (Equation 1)). It is possible to see that some connections were dropped, resulting in two different groups, which are named as A and B. For different values of $d_{max}$, it is possible to obtain different classifications of groups, as presented in Figure 11c. This result was obtained by reducing the maximum distance to $d_{max}=0.32$p.u.

### 4.2. Including Orientation

Orientation is an important feature, as it can indicate the interaction between people. Therefore, here are some results of triangulation, with the addition of orientation as a parameter of group classification. In Figure 12a, it is possible to observe that person number 3 is not part of the group (dotted blue triangle). The orientation of person 3 is opposite to the main triangle, representing here the situation of a non-interaction with the people included in triangle 1. Now, in Figure 12b, the person 3 orientation was set to point to the blue triangle, and consequently, the proposed algorithm indicates that this person is a member of this group now. Figure 12c presents a simulation result for the number of people $n_{p}=8$. Red circles represent the center of mass for each triangle; this method will be used later to draw the asymmetric Gaussian function.

### 4.3. Modified Asymmetric Gaussian with Socialization Features

After group classification, personal and group zones will be defined. Personal and group zones can be modeled using different shapes such as circular, egg shape taller on the front, ellipse and smaller on the dominant side [46]; asymmetric Gaussian functions will be used here to model all cases automatically. So, Figure 13 presents five different shapes obtained by changing parameters in the application of an asymmetric Gaussian, considering the feature values and their influence on the left, right, front and rear sides.

The concentric circles shape is the simplest one and in most works is obtained by using symmetric Gaussian functions. In this approach, it can be obtained by setting the same influence on all sides of the asymmetric Gaussian function with any value of the feature. For instance, Figure 13a shows this shape by setting the feature value as 0.5, the front influence as 0.5, the left influence as 0.5, the right influence as 0.5, and the rear influence as 0.5. It is important to remember that influences on four sides of the Gaussian functions reflect the influence of a given feature on the four sides of a person considering the frontal and sagittal axis; also, the feature value reflects a specific situation of a given feature in relation to the human being.

The egg shape is another form to model proxemics zones, and it is commonly used to model the fact that the front, left and right sides are more important when thinking about the influence of some socialization features into the socialization process. The last occurs because in socialization processes, people pay more attention to situations, objects and persons situated on those sides. Figure 13b shows how it is possible to obtain that shape in the proposed approach by setting the next values: featurevalue=0.5, InfluenceFront=−0.5, InfluenceLeft=−0.2, InfluenceRight=−0.2, and InfluenceRear=−0.1.

The ellipse shape contains certain asymmetry between the sagittal and frontal axis; however, it does not consider any asymmetry into the axis. The proposal of Rachel Kirby [47] is an example of an asymmetric Gaussian function specially conditioned to obtain ellipsoidal shapes. Figure 13c,d show two ellipse shapes; the first considers the largest influence of socialization features along the sagittal axis, and the second one considers the largest influence of the socialization features along the frontal axis. The ellipse shown in Figure 13c was obtained by setting the feature value and the influence of the socialization feature as next: featurevalue=0.5, InfluenceFront=−0.5, InfluenceLeft=0.5, InfluenceRight=0.5, and InfluenceRear=−0.5. In addition, the ellipse of Figure 13d considers the next values: featurevalue0.5, InfluenceFront=0.5, InfluenceLeft=−0.5, InfluenceRight=−0.5, and InfluenceRear=0.5.

Finally, the dominant side shape is one of the most used asymmetric shapes (i.e., standard shapes models). This shape is very important to model asymmetries caused by cultural-based social constrains. Examples of this kind of constraints include social rules to serve a dinner, where some service styles recommend serving a beverage from the right and the food from the left. In both cases of the example, the person (client or guest) has a different dominant side depending on the moment, which changes dynamically during the dinner. Figure 13e shows a dominant shape obtained with the next parameters: featurevalue=0.5, InfluenceFront=−0.5, InfluenceLeft=−0.4, InfluenceRight=−0.1, and InfluenceRear=−0.1.

In addition to testing the ability to generate various forms of proxemic zones, it is important to test how different features values will influence the size of zones around a person. So, Figure 14a shows how the changes in emotional states of a person will affect the size of the personal zone.

In the example, the emotional states presented in the table are considered in Table 1, where the more annoyed the person is, the robot perceives he/she as less friendly and therefore prefers not to approach from the front side; on the contrary, the happier the person is perceived, the more friendly the robot will perceive he/she and will prefer to approach him/her from the front side. This will cause the robot to perceive that the personal zone of the human being increases or decreases on the front side proportionally to their mood (i.e., it increases when the person is most upset and decreases when he/she is happier).

Figure 14a was obtained by setting the front influence as 0.9, the left influence as 0.5, the right influence as 0.5, and the rear influence as 0.0. In addition, the emotional state feature value was setted as 0.14 for an angry person, 0.28 for a disgusted person, 0.42 for a fearful person, 0.56 for a sad person, 0.7 for a neutral person, 0.84 for a surprised person, and 0.98 for a happy person. It is important to note that there are different values for the influence on all sides of a person, and there are also different values for features, which will cause different shapes and proximity values (i.e., personal zone sizes); so, it is important that when scientists and practitioners use this approach, they model all these values very carefully.

The present proposal has the advantage of allowing several features to be combined so that they all together influence the sizes of the proxemic zones. In this sense, this powerful property will be proved by considering an elegant dining scenario where a waiter robot is responsible for serving people. As stated before, some service styles recommend (as social rules) serving a beverage from the right and the food from the left.

As explained before, it is important to note that in both cases of the example, the person (client or guest) has a different dominant side depending of the moment, which changes dynamically during the dinner (i.e., when he/she wants food or a beverage). A simple form of understanding and implementing situations such as those explained before is by using different Gaussian functions depending on the moment. However, the proposed approach is prepared to permit developers to change dynamically sizes and forms of personal zones by combining features in a linear way, where the influence values act as weights of the feature values on each of the four sides of the person.

For testing this characteristic, two socialization features are proposed: food need and beverage need. As shown in Table 2, those features have three different values called: not need, neutral, and need; with values 0.1, 0.5, and 0.9; so, if a person needs food and a beverage, the values of both features will be 0.9, if a person does not need food but needs a beverage, the values of both features will be 0.1 and 0.9, respectively; if a person needs food but does not need a beverage, the values of both features will be 0.9 and 0.1, respectively, and so on.

Figure 14b shows how different combinations of need for food and drink (by customers or guests at dinner) affect the sizes of their personal proxemic zones, which will allow (or not) that the robot dares to approach from both the left and right sides depending on the needs of the person. That figure presents four situations: the person needs food and a beverage (Foodneed=0.9 and Beverageneed=0.9), the person is neutral about food but does not need a beverage (Foodneed=0.5 and Beverageneed=0.1), the person needs food but is neutral to beverage (Foodneed=0.9 and Beverageneed=0.5), and the person does not need food but needs a beverage (Foodneed=0.1 and Beverageneed=0.9).

The social rule defined in this example establishes that when a person needs food, the waiter robot must serve it from the left, and when the person needs a beverage, the waiter robot must serve it from the right. Then, those features will influence only the left and right side, respectively. So, the influence of both was defined as: InfluenceFront=[0,0], InfluenceRight=[0,0.5], InfluenceLeft=[0.5,0], and InfluenceRear=[0,0]. The results show that the proposed approach is well suited for combining several features for influencing the size of proxemic zones (modeled using asymmetric Gaussian functions).

### 4.4. Asymmetric Gaussian in Groups of People

Finally, it is important to prove the applicability of the proposed method in groups of persons. That means proving it by forming groups using polygonal triangulation, asymmetric Gaussian functions into groups, and socialization features to dynamically change the form and size of proxemic zones.

Figure 15 shows results of forming groups and applying asymmetric Gaussian functions to model their proxemic zones by using the emotional state socialization feature. In Figure 15, four groups are formed, and these define the personal space of the groups by setting the emotional state of the first group as sad (featurevalue=0.56), the emotional state of group 2 as neutral (featurevalue=0.7), the emotional state of group 3 as disgusted (featurevalue=0.28) and the emotional state of group 4 as happy (featurevalue=0.98). It is important to note that the group has one unique feature value which can be calculated in the perception module of the robot by combining the feature values of individuals by averaging, finding the statistical mode, and so on.

In all the groups, the influence of the feature was setted as InfluenceFront=0.6, InfluenceRight=0.4, InfluenceLeft=0.4, and InfluenceRear=0.3. However, it is possible to note that the shapes of all the Gaussian functions do not correlate as when working with individuals. The last occurs because the shape of a Gaussian function in a group (and its corresponding personal space) is not influenced only by the influence values but rather the positions of all the members of the group. Finally, the orientation of the group is defined as the mean of the orientations of persons, which is well suited for groups attending social events.

### 4.5. Path Planning and Navigation Using the Proposed Method

The full basis of the proposed method has been presented. In the previous sections, we used the triangulation method, based on Delaunay triangulation, as a simple way to connect people as vertices of triangles to create groups of people. Then, we presented the modified asymmetric Gaussian method as a way to modify the shape of the Gaussian to cover a group of people, and this shape can change based on combined features of individual people. Now, it will be presented how the proposed method can be applied for social navigation planning (see Figure 3), which is achieved by integrating triangulation and adding a Gaussian function to cover the identified groups.

In simulations, the approach proposed by Daza et al. [22], which is a method based on A* and social momentum algorithms for avoiding the personal space of persons, was used. Simulation results for the proposed method are presented in the Figure 16 and Figure 17. It is possible to see (Figure 16) that the triangulation has been applied to connect people as vertices of triangles; then, based on the proxemics and orientation of people, some connections were removed. As a result, there are three groups of people (triangles 1, 2, and 3 + 4). It is possible to see that triangles 3 and 4 are connected as a singles group, since in this case, triangle 3 is a main one ($T^{*}$), whereas triangle 4 is considered a secondary one ($T^{´}$). Based on the emotional states and positions of people, we defined some Gaussian function to create the proposed social zone.

The A* algorithm has been used to plan a path (blue line) from the robot position to the target. It is possible to see, as expected, in both results (Figure 16 and Figure 17), the resulting path planning achieved by the A* algorithm does not respect the personal space, crossing the proxemics space for groups and for a single person. On the other hand, by combining the proposed approach with social momentum (red line), it is possible to see that the calculated path is optimized from the social point of view, since the robot respects groups and individual space imposed by Gaussian functions.

The proposed method can be applied to chaotic environments, since it uses a very well-known technique (Delaunay triangulation) to connect a group of people; moreover, this proposed method can be combined with different perception, path planning and navigation algorithms for defining new features and social restrictions to provide a more suitable path planning strategy for social mobile robots, as presented in the simulation results.

## 5. Discussion

This section will discuss how the proposed method can help develop a navigation strategy closer to that performed by human beings as well as discuss how socialization features and accuracy should affect that approach.

### 5.1. Representation of Socialization Features in ROBOTS

One of the biggest challenges of robotics today is the improvement of its actions when interacting with a human being, that is, improving the human–machine interaction. However, how could a robot improve its actions considering the well-being of human beings? To answer this question, it is necessary to understand what a person normally does to be able to integrate or interact in a given social environment with other people. In this situation, most of the answers come from the characteristics perceived in an environment where people are inserted, as well as the attendance of other people. For example, the natural behavior is different when people’s emotions are considered as well as cultural and behavioral customs in a particular country or when a person interacts with a child or an adult. These environment features could help us define natural behavior to be better accepted in society.

Therefore, it is necessary to be aware of data that can be acquired from surroundings to guide better decisions. Once these features are realized, it is possible to understand and answer the question posed at the beginning of the text. To interact with people in a social context, robots need to collect and interpret data and take actions, considering social restrictions as a priority in decision making. The improvement of the representation of these features in an autonomous navigation system helps to have a greater acceptance of robots in social environments as well as a feeling of security when interacting with them.

### 5.2. Representation of Socialization Features in a Group of People

Thinking about a more realistic environment, it is possible to visualize that the robot must identify socialization features in a crowded dynamic environment, where it is possible to see that the social iterations of individuals form groups, teams, or even simple agglomerations. When this phenomenon happens, the individual features of people are mixed, and together, they will generate a new group feature. Robots that interact in public places have to be prepared to interact with groups of people and understand their main features. So, what would be the best way to validate and work on group perception in a robot? At this point, three approaches that generate discussion on this problem can be emphasized that have not been fully standardized. The first approach would be how to define the best methodology to be able to extract the main socialization features of the group of people in dynamic environments. The second question is related to the type of socialization features that will be used in the perception and what will be the degree of importance of each one of them in the robot’s decision making. Finally, how should a group of people be defined? In the article, triangulation between people who are interacting to form a group was used, for example.

### 5.3. Benefits of Social Perceptions

It is evident that if a better perception of the environment is used, it is possible to have a better social interaction with people. For this reason, it is necessary to define behaviors, actions, and attitudes in a robot, depending on the environment where it is inserted, improving the interaction of robots with humans in that social environment. Robotics is present in our lives; for this reason, it is also necessary to relate and integrate robotic perception systems into robot decision-making systems, so it is possible to have a good interaction with these machines and have them more accepted in society. Robots should learn from their actions seeking a bio-inspiration in human spatial memory, where the environment can be divided into a dynamic part and a static part, where the robot could store a memory in the static part, thus facilitating the execution of repetitive tasks.

### 5.4. Other Possible Applications for Group Gaussian

Group Gaussian can also be applied in moving groups, robot swarms, crowd shifts, riot control, dance, and choreography. An example is an opening of a sporting event, where a group of people moves at an orderly pace to form figures or symbols, as in the case of the opening of the Olympic Games. The Gaussian would give this moving group the lightness of an individual, making it easier to calculate the group’s velocity and quantity of motion, where everything would be calculated in relation to the group’s center of mass.

### 5.5. Accuracy of Weights/Influences of Socialization Features in Robots

For a robot to safely and efficiently navigate a dynamic and congested environment, it has to overcome several obstacles, for example, the inaccuracy inherent in its perception of the environment, obtained by sensors noise or errors, sensor capacity limitations or even the dynamism of the environment. Another source of uncertainty for robots is noise or errors from their actuators or effects. Another situation is when the robot has to mimic human behavioral features and these features are not discrete, even among humans; depending on the degree of education, geolocation, gender, age, etc., certain features can weigh more or less in the decision making. Thus, a mathematical model that includes the uncertainties of both sensors and actuators and the weights and influences of socialization features is necessary to define correct robotic navigation with social restrictions and limitation hardware.

## 6. Conclusions

This paper has proposed a new approach for including multiple social constraints in robot social navigation. The proposed method can be applied to a real-world perspective, considering a social context and interactions between people, even in a crowded environment.

First, we proposed a new form of perceiving groups of persons in social space by using Delaunay triangulation, which in an incremental way (starting with groups of three persons) performs the recognition of groups of persons by its orientation and distance. Individuals and recognized groups have their proxemic zones modeled by asymmetric Gaussian functions. The proposed approach permits the recognition of groups in logarithmic time because of its nature of calculating it in an incrementally way considering a triangulation process.

We defined as a complete asymmetric Gaussian function based on the sagittal and frontal axis of persons and groups. Individual orientation must be obtained from the perception module of a robotic system, and group orientation must be calculated by using a mathematical function of orientations of the persons that are part of the group. In the proposed experiments, the mean of orientations of persons in a group, which is acceptable for persons attending social events, was used.

We also proposed the use of numerical vectors called *socialization features* to model the perceived openness to an interaction between humans and a group of humans. Those features have the capability of modifying the size and form of the proxemic zones. In the simulation results, it was presented how those features (in the individual and combined ways) can modify the size and form of personal space in order to permit robots to approach to humans or groups or not depending on situations. In addition, a linear combination of features triggered by *influence values* on the four sides of a person (i.e., its Gaussian function) models the reaction of the robot to the value of the feature (negative to move away from human and positive for approaching the human). It is recommended in future work to develop new forms of combining several features.

Finally, the proposed approach was tested in path planning and navigation by using the approach proposed by Daza et al. [22], which is a method based on A* and social momentum algorithms for avoiding the personal space of persons. It demonstrated the applicability of the presented approach for path planning and navigation with optimal results and also it is recommended to use it in other approaches for path planning and navigation.

Different experiments and results are explained in this work; for example, in Figure 17, it is possible to see the final definition of a trajectory considering social restrictions. This result demonstrates that the proposal allows us to automatically include multiple social restrictions (as happens in the real world) in the interactions of robots with people and groups of them, modifying the shape and size of the proxemic areas, so that the algorithms for path planning and navigation can be easily conditioned in favor of navigation that considers various situations that occur in a combined way in the real environments of social robotics.

## Figures and Tables

**Figure 1 sensors-22-04602-f001:**
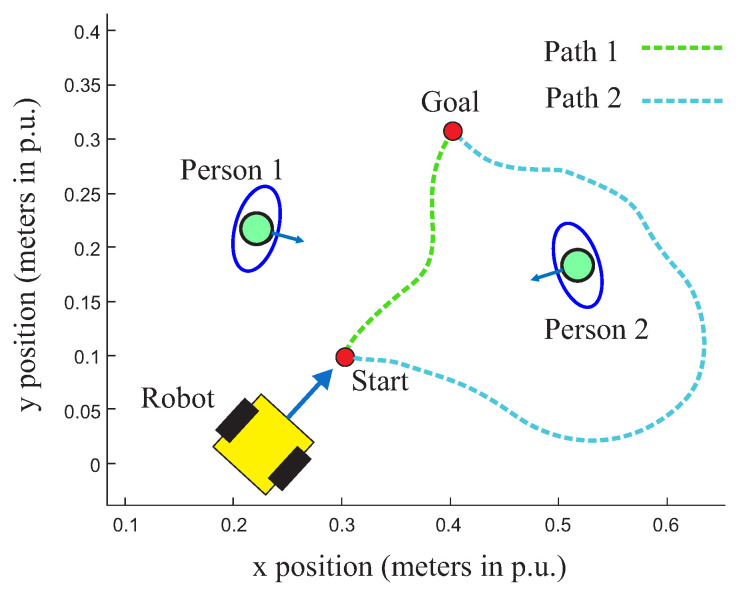
Path planning without social conventions (green line) and with social constraints (blue line).

**Figure 2 sensors-22-04602-f002:**
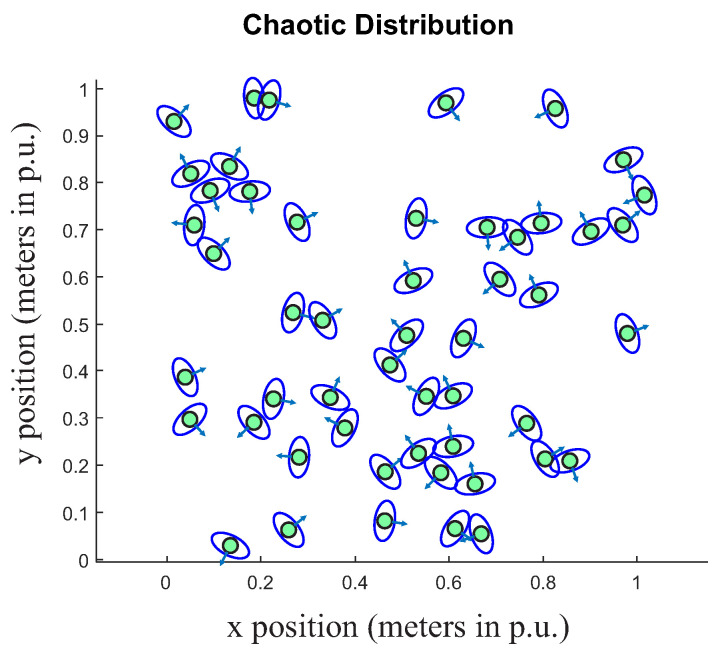
Example of a chaotic distribution of people. Number of points n=50.

**Figure 3 sensors-22-04602-f003:**
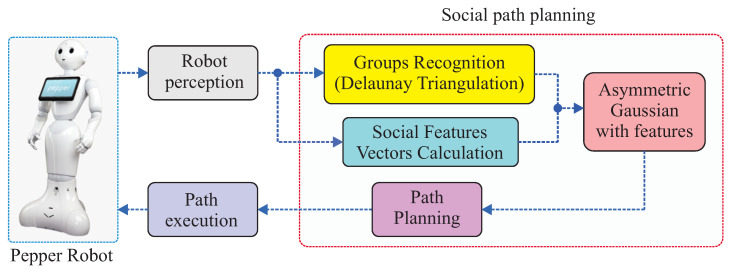
Social path planning using the proposed method.

**Figure 4 sensors-22-04602-f004:**
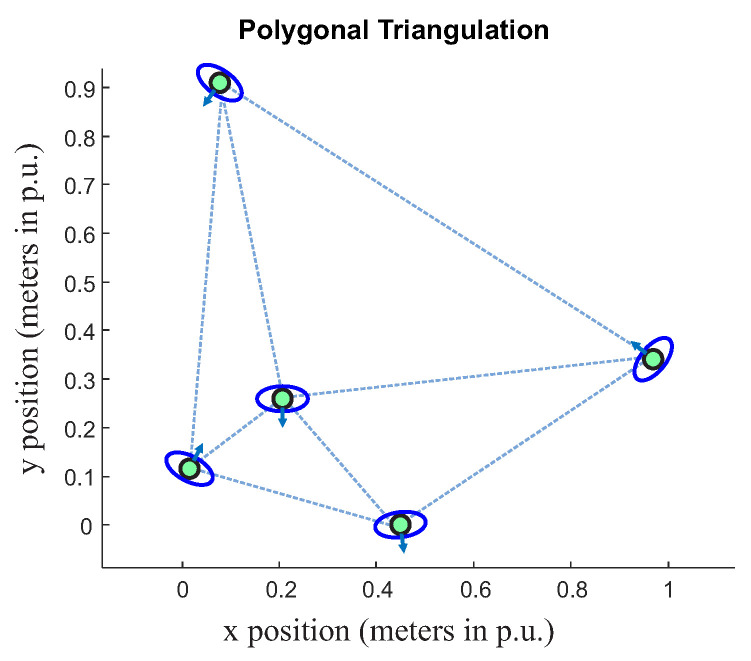
Example of application of Delaunay triangulation. Number of points n=5.

**Figure 5 sensors-22-04602-f005:**
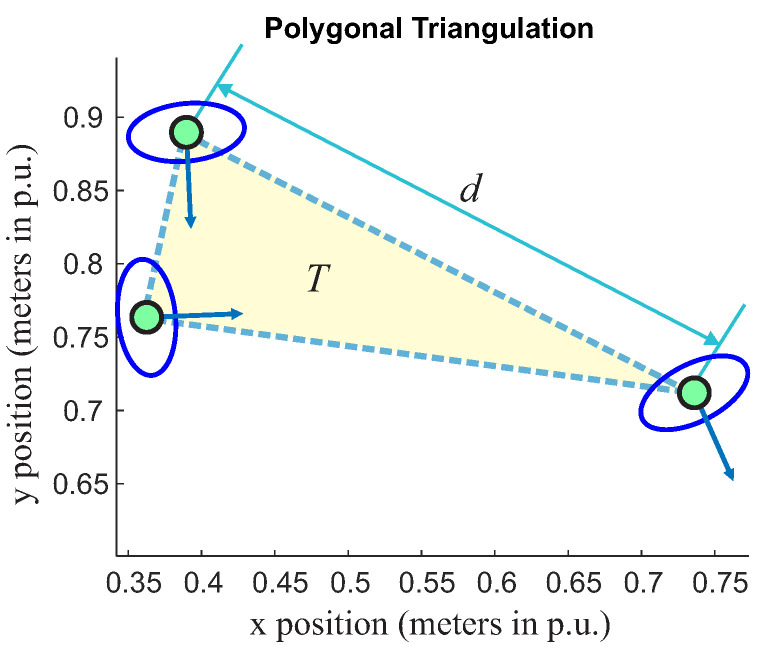
Distance between vertices. The yellow triangle is related to *T-Space*.

**Figure 6 sensors-22-04602-f006:**
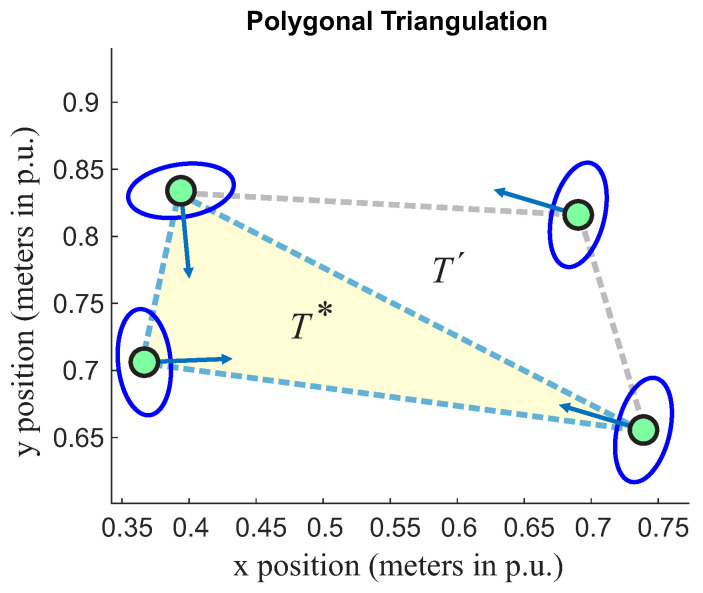
Group classification. The yellow triangle is related here to the reference triangle ($T^{*}$) for group classification.

**Figure 7 sensors-22-04602-f007:**
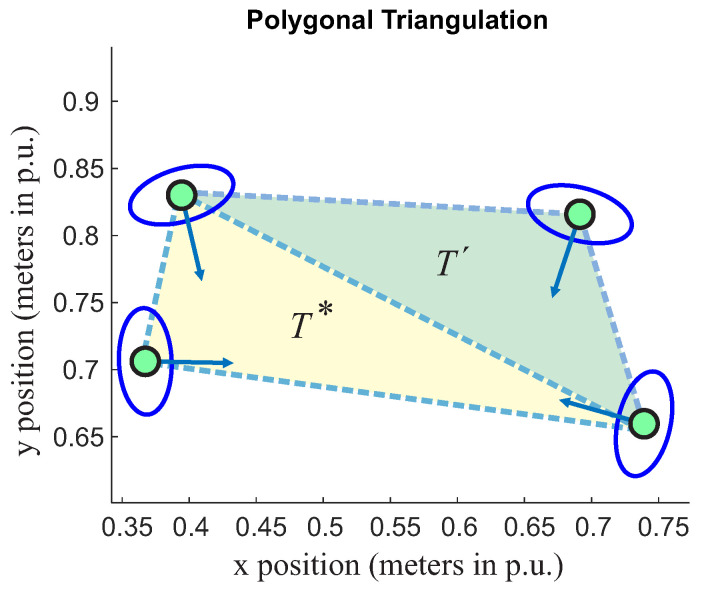
Group classification including a peripheral triangle. The yellow triangle is related here to the reference ($T^{*}$), while the green triangle is the peripheral one ($T^{{}^{\prime }}$) for group classification.

**Figure 8 sensors-22-04602-f008:**
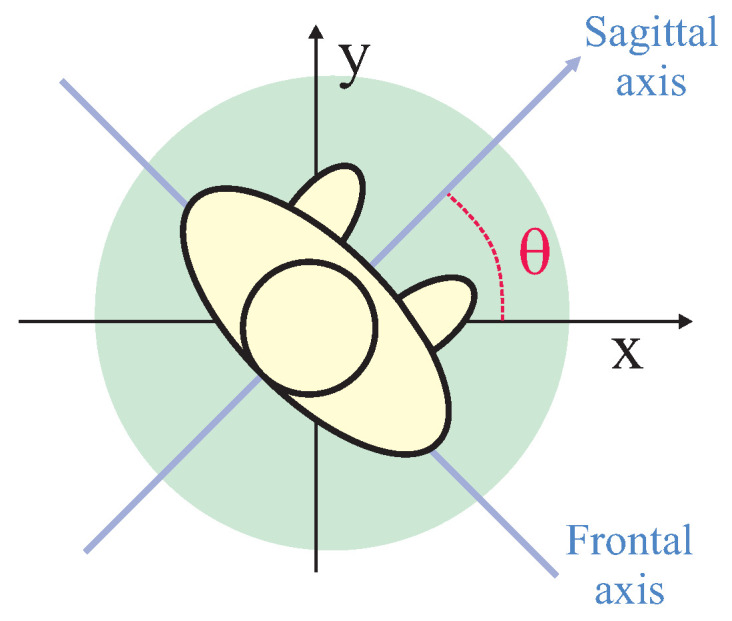
Basis of Asymmetric Gaussian Calculus.

**Figure 9 sensors-22-04602-f009:**
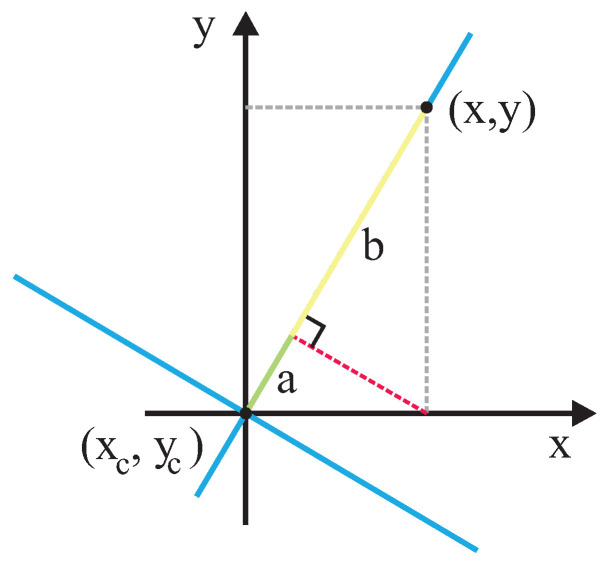
Calculus of distance in sagittal axis for a given (*x*,*y*) point in the reference frame.

**Figure 10 sensors-22-04602-f010:**
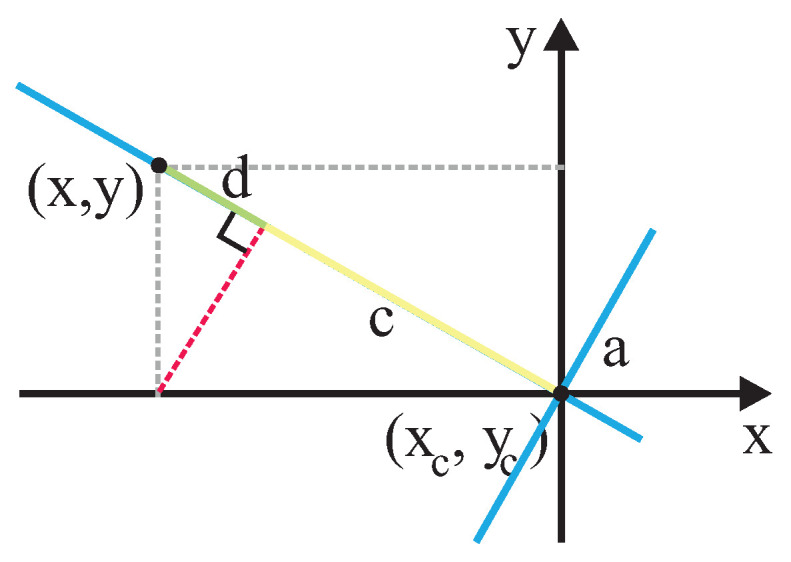
Calculus of distance in frontal axis for a given (*x*,*y*) point in the reference frame.

**Figure 11 sensors-22-04602-f011:**
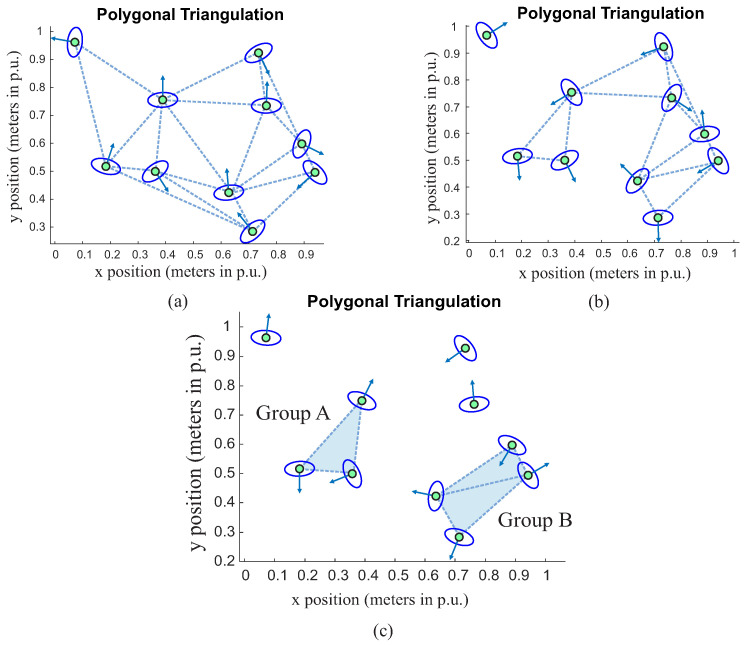
Group classification by using Delaunay triangulation. (**a**) Without limit of distance, (**b**) with distance limitation $d_{max}=0.40$
p.u. and (**c**) with distance limitation $d_{max}=0.32$
p.u. Blue triangles mean the resulting groups (Group A and B).

**Figure 12 sensors-22-04602-f012:**
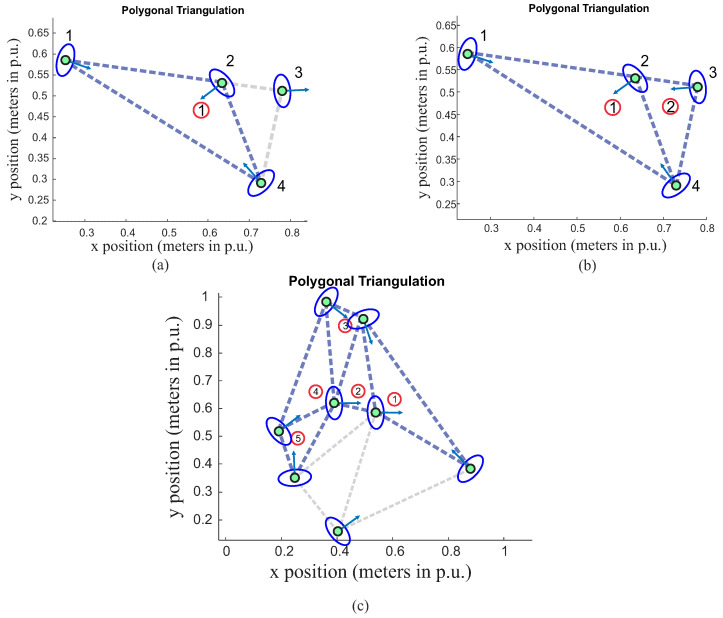
Social group classification: (**a**) number of people $n_{p}=4$ with person exclusion, (**b**) number of people $n_{p}=4$ with person inclusion, (**c**) number of people $n_{p}=8$. Red circles are related to calculated center mass of each triangle, while number 1 in the red circles refers to reference triangles ($T^{*}$) and numbers 2, 3, …, 5 are referred to peripherals $T^{{}^{\prime }}$ ones.

**Figure 13 sensors-22-04602-f013:**
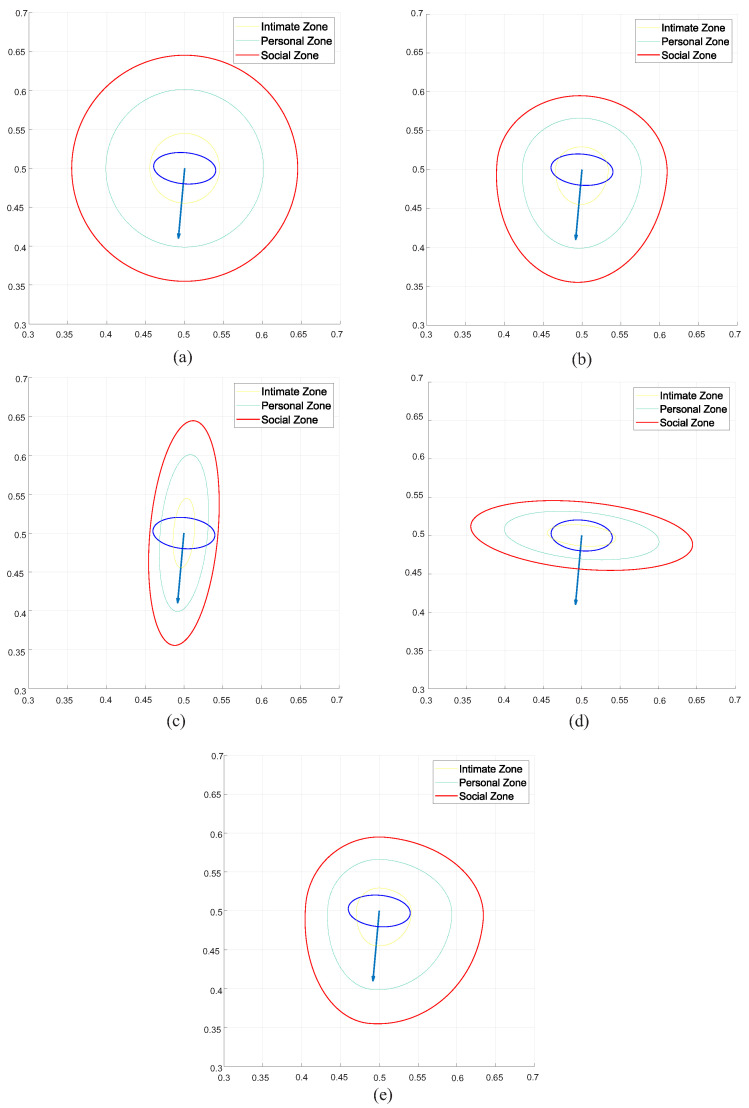
Gaussian Shapes: (**a**) Concentric Circles, (**b**) Egg Shape, (**c**) Ellipse Shape, (**d**) Ellipse Shape, and (**e**) Dominant Side.

**Figure 14 sensors-22-04602-f014:**
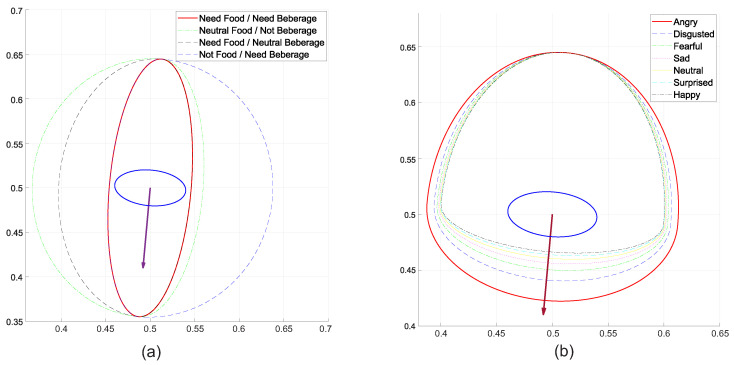
Emotional State: (**a**) Influence of Emotional States into Personal Zone Size; (**b**) Personal Space Influence of Two Features for a Waiter Robot: Diner Food Need and Beverage Need.

**Figure 15 sensors-22-04602-f015:**
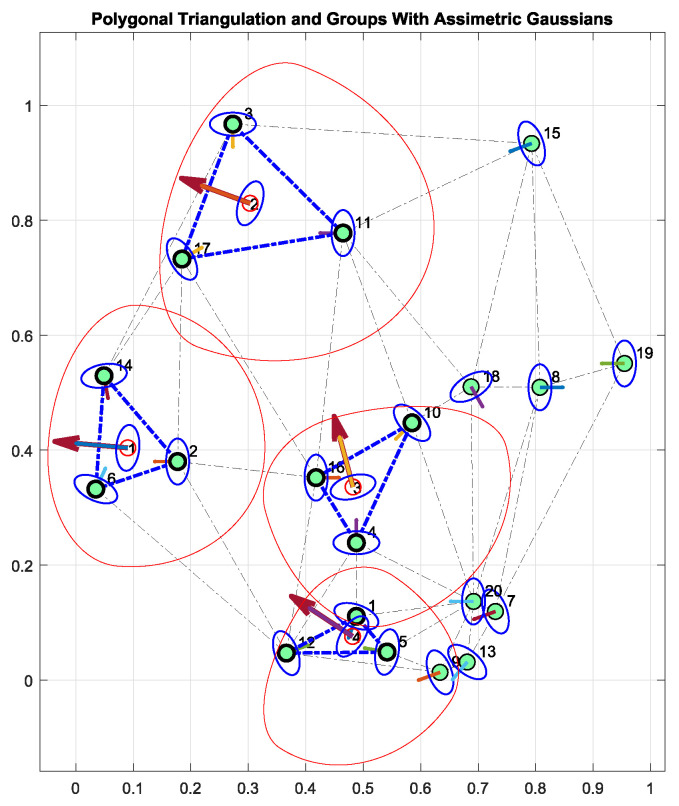
Influence of Emotional States into Groups. Red color refers to the proposed group Gaussian function for group space.

**Figure 16 sensors-22-04602-f016:**
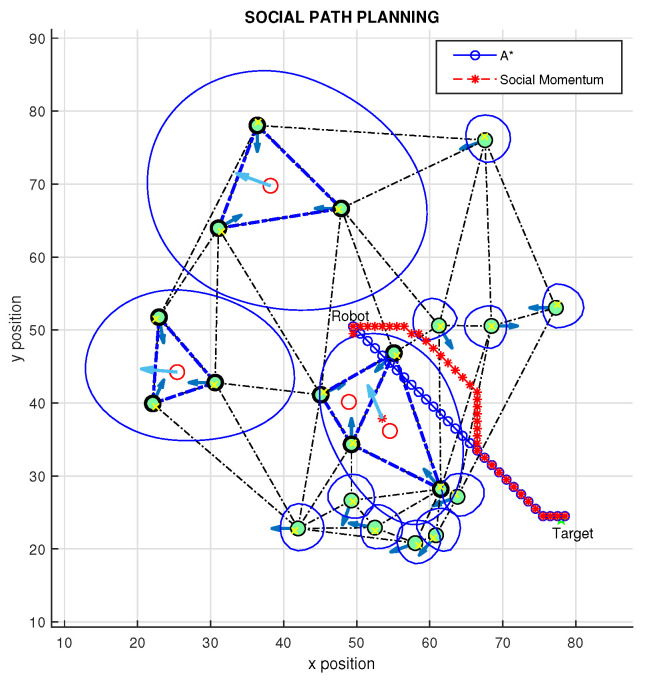
Proposed path planning method.

**Figure 17 sensors-22-04602-f017:**
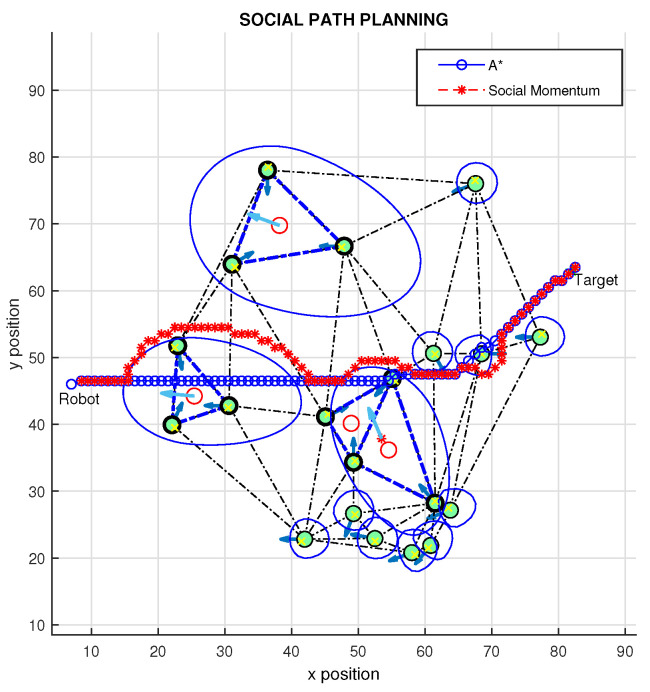
Proposed path planning method, different target.

**Table 1 sensors-22-04602-t001:** Values for socialization feature “emotional state”.

Emotional State	Feature Value $v_{j}$
Angry	0.14
Disgusted	0.28
Fearful	0.42
Sad	0.56
Neutral	0.70
Surprised	0.84
Happy	0.98

**Table 2 sensors-22-04602-t002:** Values for Socialization Features in a Dinner: “Food Need” and “Beverage Need”.

Food Need	Value $v_{j}$	Beverage Need	Value $v_{j}$
Not Need	0.1	Not Need	0.1
Neutral	0.5	Neutral	0.5
Need	0.9	Need	0.9

## Data Availability

Not applicable.

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
