# Peer review of "A New Approach for Including Social Conventions into Social Robots Navigation by Using Polygonal Triangulation and Group Asymmetric Gaussian Functions"

_sensors, 2022, doi:10.3390/s22124602_

Round 1

Reviewer 1 Report

The authors propose an improved method (asymmetric Gaussian functions modified dynamically according to socialization features having different influence on each side) for taking into account more accurately the social conventions (apart from person position and orientation in each group of people – improved polygonal triangulation) and thus to achieve more “polite” robot path planning, in dynamic environments with many persons forming different groups. 

The paper provides interesting and multi-perspective information enhancing previous methods, plus useful evaluation/simulation results. Nevertheless, some parts should be further improved and some issues should be better addressed. The following reviewing suggestions are based on the second download of the manuscript, i.e., the one with the missing references corrected:

1)  In the introduction section, referring to the general scientific background – lines 30-31, the authors may find beneficial to include more material dealing with service robots definitions/taxonomy and examples, like the ones contained into the following research:

Current Trends: https://doi.org/10.3390/app112210702, Delivery: https://doi.org/10.3390/electronics11050748,  Catering: https://doi.org/10.3390/electronics10010047,  Logistics: https://doi.org/10.1016/j.ejor.2021.01.019, Agriculture: https://doi.org/10.3390/machines9040082, Surveillance: https://doi.org/10.3390/s20041097, Rescue: https://doi.org/10.1002/rob.21887, Tourism:  https://doi.org/10.3390/tourhosp2010003

2) Some English language grammar and style issues should be corrected. Some formatting issues should also be corrected.

3) The authors are encouraged to provide all the necessary references for the algorithms they mention, as for instance for the heuristics D* and A* used for path planning.

4) Sections 2 and 3 have the same title and quite similar content, this issue should be addressed in the updated manuscript version.

5) Update title for Figure 3 properly, as both the “impolite” and the “polite” paths are shown.

6) Explain unit percentage (p.u.) much earlier than in line 565.  The m unit for d should be omitted, as p.u. does not contain units (e.g., meters).

7) The authors should better explain how the step 3, as described in lines Ln533-536, was implemented according to the proposed approach. The relevant explanation, in lines 680-702, should be enriched with more details on the methods used for calculating the group characteristics and functions from the individuals participating in each group, and even better, a more analytic form of it should be placed earlier into the manuscript.

8) The authors provide few information on the experimental setup, the computer that hosted the experiments, the platform and its configuration, and the sensor data being used in the simulation for verifying the performance of their proposed approach. Further linking with real platform paradigm (sensing/perception/path planning) should also be welcome.      

9) In section 6, before subsections, an introductory paragraph would be beneficial.

10) Some reassessment of the Conclusions section, in order to be more compact and to better summarize the whole work being presented, should be beneficial.  

Reviewer 2 Report

The manuscript is devoted to the problem of the robot path planning in a social environment. The work contains original ideas, undoubtedly of interest. Authors propose taking into account the individual and group behavior of people presenting the obstacles to the movement of the robot, as well as their emotional state, which determines the degree of danger of such an individual from the point of view of a possible collision.

However, unfortunately, the work cannot be accepted in its current form.

The manuscript is poorly structured. Instead of a clear division into Introduction, Materials and Methods, Results and Conclusion & Discussion Sections, authors propose their own structure such that e.g. the novelty of the proposed approach appears only in the 4-th section, not in the Introduction, as it should be. As a result, the manuscript, in my opinion, is not easy to follow, and also it is too long (27 pages). The aim of the work is described vague and should be reformulated.

Abstract is also too long. Conclusion contains no numerical evaluations of the obtained results.

A critical error is in the citations, as well as references to figures - question marks are instead of numbers. Also, there is no bibliography.

As a final remark, I do not think that the article should be published in the journal Sensors, because its contribution is not directly related to measurements and sensory data processing. Instead, it assumes that the data on the location of people in space has already been obtained. I think that the paper after major revision is suitable to be published in Mathematics, Algorithms, Robotics and other related journals.

Round 2

Reviewer 1 Report

The authors made several improvements in their updated version, trying to be in line with the suggestions of the reviewers. Nevertheless, reassessment is necessary:

-  The numerous changes in both the content and the structure of the article demand meticulous reworking of the whole manuscript.  

-  Further elaboration in terms of English language grammar and style is required, especially for the newly added parts, while the use of 1st and 2nd person in the description should be avoided.

- The fact that most of the work has been performed on a simulation environment should be clarified much earlier in the paper (i.e., apart from the beginning of section 4).

- The declaration about 100% accuracy should be replaced with a more modest statement about the performance of the proposed method, preferably at the results section.        

Reviewer 2 Report

Dear Authors!

The current version of the manuscript is a significant improvement over the first version. The objectives of the study and your contribution have become clear. The structure of the article has been corrected in favor of better ease of following. The abstract and introduction have been substantially revised, and now provide the reader with the necessary information. The bibliography contains an impressive 48 references. Mathematical expressions for the asymmetric Gaussian function are given, the steps of the developed algorithms are described.

I may note only a few shortcomings.

1. In Figure 14, please enlarge the legend.

2. While listing contributions, use the uniform style of upper- and lowercase itemizing.

3. In Figure 13, there is no color legend, need fixing.

There may be some other design and grammar flaws, please, inspect the manuscript properly.

Taking into account the high level of the research and presentation, I believe that the article can now be accepted after minor revisions.
